# Spread Layers of Lysozyme Microgel at Liquid Surface

**DOI:** 10.3390/polym14193979

**Published:** 2022-09-23

**Authors:** Olga Yu. Milyaeva, Alexander V. Akentiev, Alexey G. Bykov, Shi-Yow Lin, Giuseppe Loglio, Reinhard Miller, Alexander V. Michailov, Ksenia Yu. Rotanova, Boris A. Noskov

**Affiliations:** 1Institute of Chemistry, St. Petersburg State University, Universitetsky pr. 26, St. Petersburg 198504, Russia; 2Chemical Engineering Department, National Taiwan University of Science and Technology, Taipei 106, Taiwan; 3Institute of Condensed Matter Chemistry and Technologies for Energy, 16149 Genoa, Italy; 4Physics Department, Technical University of Darmstadt, 64289 Darmstadt, Germany

**Keywords:** β-lactoglobulin, lysozyme, microgel particles, spread layers, IRRAS, BAM, AFM, SEM, surface dilational viscoelasticity

## Abstract

The spread layers of lysozyme (LYS) microgel particles were studied by surface dilational rheology, infrared reflection–absorption spectra, Brewster angle microscopy, atomic force microscopy, and scanning electron microscopy. It is shown that the properties of LYS microgel layers differ significantly from those of ß-lactoglobulin (BLG) microgel layers. In the latter case, the spread protein layer is mainly a monolayer, and the interactions between particles lead to the increase in the dynamic surface elasticity by up to 140 mN/m. In contrast, the dynamic elasticity of the LYS microgel layer does not exceed the values for pure protein layers. The compression isotherms also do not exhibit specific features of the layer collapse that are characteristic for the layers of BLG aggregates. LYS aggregates form trough three-dimensional clusters directly during the spreading process, and protein spherulites do not spread further along the interface. As a result, the liquid surface contains large, almost empty regions and some patches of high local concentration of the microgel particles.

## 1. Introduction

The application of protein aggregates in food, pharmaceutical, and cosmetic branches of industry, in particular to create foams and emulsions of high stability, resulted in recent years in numerous publications on surface properties of aggregate dispersions [1,2,3,4,5,6,7,8,9,10,11,12,13,14,15,16,17,18,19,20,21,22,23,24,25,26,27,28,29,30,31,32,33,34,35,36,37], mainly of dispersions of β-lactoglobulin (BLG) [2,5,12,13,14,15,21,27,30,32,33,34,37] or whey protein [1,4,19,26] aggregates. It turns out that, in spite of the fact that the protein aggregates are better emulsifying and foaming agents than native proteins, the surface properties of their dispersions are rather close to those of the corresponding native proteins [5,6,7,8,9,12,13,14]. This similarity in surface properties is caused partly by difficulties in the purification of these dispersions, which can contain some polypeptides of relatively low molecular weight in addition to the aggregates and presumably protein monomers. The polypeptides are produced as a result of the protein hydrolysis at high temperature in the course of aggregate formation, and are characterized by a significant surface activity and a higher adsorption rate compared with the protein aggregates [5,8,9,13,30]. Therefore, the ratio of local concentrations of polypeptides and protein aggregates in the surface layer proves to be much higher than in the bulk phase, and the polypeptides strongly influence the surface properties of protein aggregate dispersions. The careful purification of the dispersions does not allow excluding the influence of polypeptides because of their high surface activity. Another possible cause of the high polypeptide concentration in the surface layer is their release from protein aggregates in aqueous dispersions [38,39].

Among various protein aggregates, fibrils and dense compact protein nanoparticles have been studied to the most extent [1,2,4,9,11,13,16,17,24,32,33,37]. The latter usually consist of interconnected and partially unfolded protein chains and are known as protein microgel [4,9,27,28]. The LYS and BLG microgels also have an almost spherical shape and, therefore, protein microgel and protein spherulite are used as synonyms below.

It was shown recently that the layers of BLG aggregates at the water surface, which are produced by spreading a concentrated aqueous dispersion on an aqueous subphase, can contain lower relative concentrations of polypeptides. As a result, the properties of the spread layers of BLG microgel dispersions differ strongly from those of adsorption layers of the same particles, allowing an estimation of the influence of the protein aggregate morphology, and interactions between the aggregates, on the properties of their layers on an aqueous subphase [30]. Although the size and morphology of protein particles can influence the dynamic properties of their layers at the water–air interface [26], the observed differences can be much lower than those between spread and adsorbed layers of the same BLG particles. To the best of our knowledge, there is no similar information on the distinctions between the properties of spread and adsorbed layers of aggregates of other globular proteins and on the influence of the protein structure on the properties of the protein aggregate layers on a water surface. Therefore, this study is devoted to spread layers of microgel particles of another well-investigated protein, lysozyme (LYS), and the obtained results are compared with the corresponding data for adsorbed layers of this microgel and with those for spread and adsorbed layers of BLG microgel, which have been studied before [30].

## 2. Materials and Methods

LYS (Sigma-Aldrich, St. Louis, MO, USA, M_w_ ≈ 14,300 Da) and BLG (Sigma-Aldrich, USA, M_w_ ≈ 18,300 Da) were carefully purified before the microgel preparation. Triply distilled water was used to prepare 10 wt.% protein solutions. The solution pH was adjusted to 10 by the addition of small amounts of 0.1 M NaOH. After that, the solution was dialyzed against water at pH 10 for three days using a cellulose membrane (Sigma-Aldrich, Schnelldorf, Germany). The protein concentration after dialysis was 2 wt.%. To remove undissolved substances, the solution was centrifugated (10,000× *g*, 20 min) and filtered through a membrane with a pore size of 200 nm (Vladipore membranes, Vladimir, Russia).

The flask with the purified LYS solution was placed into a hot-water bath at 90 °C and heated under rotation for 20–300 min until it became muddy. After that, the flask was immediately cooled in an ice-water bath, and the obtained microgel dispersion stored in a refrigerator at 4 °C.

In order to remove nonreacted protein and peptides, the dispersion was centrifuged (at 5000× *g* for 15 min.) and the supernatant was replaced by water. Shaking a test tube allowed resuspension of microgel particles in water. The whole purification procedure required up to 3 exchange steps.

To spread LYS microgel layers, the dispersion was added drop by drop to a liquid subphase using an inclined glass plate, which was partially immersed into the liquid. The subphase was 0.1 M NaCl solution at pH 10, if not otherwise noted. In this case, the high ionic strength and the proximity to the LYS isoelectric point decreases the influence of electrostatic effects. High elasticity of BLG spread layers was obtained previously under similar conditions [30]. In some cases, as described below, the addition of up to 20% of ethanol to the spreading dispersion improves the spreading process and means highly homogeneous layers of LYS particles are obtained.

Similar procedures, as described in detail previously, were used to obtain BLG microgel and its layers on aqueous subphases [30].

The dynamic surface elasticity and surface tension were measured using an ISR instrument equipped with a Wilhelmy plate (KSV NIMA). The accuracy of the surface tension measurements was ±0.2 mN/m.

The instrument was equipped with two barriers, and their motion with reversion in opposite phases led to sinusoidal oscillations of the surface area [40,41]. If the relative amplitude of the surface area oscillations is much less than unity, the system response to sinusoidal deformations is linear and one can define the dynamic surface elasticity:(1)$$E\left(\omega \right)=E_{r}+iE_{i}=\frac{\delta \gamma }{\delta lnA}$$
where *γ* is the surface tension; *A* is the surface area; *ω* is the angular frequency; and *E*, *δγ*, and *δA* are complex quantities.

All surface rheology measurements in this work were performed at a constant frequency of 0.03 Hz and oscillation amplitudes of 5%. The accuracy of the dynamic surface elasticity was close to ±3%. The imaginary part of the dynamic surface elasticity was always much less than the real part, and, therefore, only the modulus of the surface elasticity is discussed below.

The macro- and micromorphology of the layers were characterized by Brewster angle microscopy (BAM) (BAM 1, Nanofilm Technology, Goettingen, Germany), scanning electron microscopy (SEM) (Zeiss Merlin, Aalen, Germany), and atomic force microscopy (AFM) (NTEGRA Prima setup, NT-MDT, Moscow, Russia). The microgel layers were transferred from the liquid surface onto the surface of a freshly cleaved mica plate (for AFM) or an atomically smooth silica plate (for SEM) by the Langmuir–Schaefer method. Each sample was dried in a desiccator at room temperature for 3–5 days. The AFM measurements were carried out in the semi-contact mode. The SEM measurements were performed at an operating voltage of 2 kV and different magnifications.

The infrared reflection–absorption spectra (IRRAS) were recorded using a Nicolet 8700 FTIR spectrometer (Thermo Scientific, Waltham, MA, USA). The instrument was equipped with a tabletop optical module (TOM) described elsewhere [42].

The IR single beam spectra were registered with a MCT-D detector. The IR beam was focused on the surface of a sample using a BaF2 lens with a focal length of 750 mm. A linear wire grid polarizer mounted on a motorized rotary translator was used to form polarized IR radiation. The polarizer positioning accuracy was about 0.1 degree. The spectrometer and TOM were purged with nitrogen. In all spectra measurements, 2048 scans were accumulated with a resolution of 4 cm^−1^. The data were collected at an angle of incidence equal to 40°. IRRAS data were presented as plots of reflectance–absorbance (RA) versus wavenumber:(2)RA = −log R/Rb,
where R and Rb are the reflectivities from solutions with a spread layer and pure buffer solutions, respectively. The alignment of the equipment was checked according to the procedure described earlier [42].

## 3. Results and Discussion

The protocol of preparation of LYS microgel dispersions differs from that of BLG microgel, and the obtained LYS aggregates are characterized by a higher level of polydispersity than that of BLG aggregates. The AFM image of LYS microgel aggregates was obtained after drying a dispersion drop on the surface of a freshly cleaved mica plate (Figure 1). The almost spherical aggregates have diameters in the range from 150 to 600 nm.

The dynamic elasticity of spread layers of LYS spherical aggregates depends on the purification degree of the dispersion, as in the case of the layers of BLG spherulites, but, unlike the latter case, the layer’s elasticity proves to be always less than the values for spread layers of the native protein. The elasticity does not increase noticeably with the increasing number of purification cycles (Figure 2).

Moreover, after three purification cycles, the spreading of their dispersion with a concentration of 10 mg/mL onto water and onto 0.1 M NaCl solution at pH 10 with a surface area of 150 cm^2^ does not lead to any changes in the surface pressure and surface elasticity, as long as less than 3 mL of the dispersion is spread. Note that the dispersion spreading onto pure water, 0.05, and 0.1 M NaCl solutions and phosphate buffer at pH 7 leads to almost the same results. The absence of noticeable changes in the surface properties in this case can be connected with the peculiarities of the spreading process, when the protein aggregates are trapped by the liquid flow and remain in the subphase without adsorption at the liquid surface. Another explanation of the negligible influence of the spreading on surface properties is the formation of dense three-dimensional structures of the protein aggregates on the liquid surface, which can coexist with almost empty regions of the liquid surface with a small aggregate surface concentration. In this case, the spread protein microgel does not propagate along the liquid surface, forms compact three-dimensional islands at the surface, and, consequently, the surface pressure remains close to zero.

On the contrary, the spreading of native LYS from a concentrated solution onto the surface of an aqueous subphase proves to be more effective, and does not lead to the formation of three-dimensional structures at the liquid surface. The spreading of 1 mL of a LYS solution with a concentration of 10 mg/mL onto the area of 150 cm^2^ is sufficient to increase the surface pressure up to 52 mN/m (Figure 2). When spreading a BLG spherulite dispersion with a concentration of 8 mg/mL onto the surface of 150 cm^2^ of 0.1 M NaCl solution, the spherulites also mainly remain at the liquid surface, and the spreading of 3 mL is sufficient to increase the surface elasticity up to 140 mN/m [30].

The application of IRRAS shows that some LYS aggregates remain at the surface of water and 0.1 M NaCl solutions after spreading the LYS microgel dispersion. The spreading results in the appearance of two amide III bands close to 1250 cm^−1^ and 1380 cm^−1^, and two broad amide II and amide I bands approximately at 1550 cm^−1^ and 1650 cm^−1^, respectively (Figure 3).

All bands arise as a result of the peptide bond vibrations in the LYS layer [43]. The intensity of these bands depends on the subphase, and very strongly on the point where the spectrometer laser beam is reflected from the liquid surface. The latter effect means that the liquid surface is highly heterogeneous. The intensity of the amide bands can exceed approximately four times the corresponding intensity for a native LYS monolayer, or approach zero within error limits if the reflection point changes (Figure 3). These results indicate that there are some spots of LYS three-dimensional structures (multilayers) on the liquid surface, while other parts of the liquid surface are almost empty and do not contain protein molecules. The intensity of the amide bands shows little change for a few hours, indicating that the protein spherulites are strongly bound in the multilayers and do not spread further along the liquid surface. Note that the spread BLG layer proves to be more homogeneous, and the scanning along the surface leads only to small changes of the amide band intensity close to error limits [30].

The surface concentration of LYS microgel depends also on the ionic strength of the subphase. The intensity of the amide bands decreases with a decrease in the ionic strength for the same spreading volumes (Figure 3 and Appendix A). In this case, the dissociation of the protein amino group increases, leading to the increase in the molecule charge. Hence, the formation of rigid surface aggregates is hindered, due to the increased electrostatic repulsion between the protein molecules.

The formation of a LYS aggregate layer on the surface of 0.1 NaCl solutions, in spite of the almost zero surface pressure, is also confirmed by an increase in π after a strong surface compression. The slope of the compression isotherm of a spread layer of unpurified LYS spherulites almost coincides with that for a native LYS layer at relatively small compressions, but increases noticeably if A/A_0_ < ~0.4 (Figure 4). In the latter case, the surface concentration of spherulites increases and the repulsion between them starts to contribute to the surface pressure. The slope of the compression isotherm of a layer of purified spherulites is higher at the beginning of the compression process from zero surface pressures and increases further at A/A_0_ < ~0.4, due to the increased repulsion between protein spherulites. Note that, unlike the compression isotherms of BLG aggregate layers, all isotherms for LYS aggregates are monotonic without surface pressure fluctuations, which appear in the case of a layer collapse [30]. Therefore, there are no abrupt transitions of LYS aggregate layers from a spherulite monolayer to the formation of three-dimensional aggregates on the liquid surface. The three-dimensional structures in the LYS aggregate layers are presumably formed on the liquid surface just after spreading, and preserved there even if the surface pressure equals zero.

The AFM images confirm the conclusions from the analysis of surface pressure isotherms and surface rheology data. The AFM images of spread aggregate layers do not show a continuous surface film. If the microgel dispersion is not purified, it is possible to observe only separate spherulites and some clusters of them inside a smoother layer (Figure 5a). This layer of lower roughness presumably is formed by polypeptides of low-molecular-weight and some protein monomers, and provides a non-zero surface pressure. If the protein aggregate dispersion is purified three times by centrifugation and supernatant exchange by an aqueous phase, some AFM images show rather close-packed structures of spherulite multilayers, while other images display some regions of low surface density with a few spherulites, or small clusters of them (Figure 5b,c). The number of these regions decreases together with the corresponding surface area, but they do not entirely disappear even at high compressions.

Therefore, the spreading of a purified dispersion of protein microgel onto the surface of 0.1 M NaCl solution at pH 10 results in the formation of tough clusters of the microgel particles with a thickness exceeding the particle diameter. These clusters are characterized by strong cohesion between the spherulites and do not disintegrate for several hours. They do not effuse single spherulites to the surrounding liquid surface and the surface remains strongly heterogeneous. The interaction between these aggregate clusters during surface compression results in a strong increase in the surface pressure. These clusters can have a mesoscopic size and can be observed by BAM even at zero surface pressure (Figure 6A–C). The layer remains almost the same after applying a mechanical disturbance (Figure 6C,D). After compression, the images become uniformly gray (Figure 6D). However, the layer is still not homogeneous. By touching the surface with a thin needle, one can observe the motion of individual microgel clusters along the surface. This behavior differs from that observed for spread layers of BLG microgel when the layer is approximately homogeneous at mesoscale, and the layer destruction (collapse) occurs only at high surface pressures.

Although the size and shape of BLG and LYS spherulites are approximately the same, the properties of their spread layers at the liquid–gas interface are different, and presumably determined by the protein primary and secondary structures (Figure 7). These findings set a fundamental task to elucidate the interrelations between the protein structure and the properties of their aggregates and layers, and indicate the necessity to study the layers of other proteins.

The morphology of the protein aggregate layers can be altered not only by changing the protein, but also by changing the spreading solvent and subphase. The addition of 20 wt.% of ethanol to an aqueous protein microgel dispersion, and using phosphate buffer as substrate, strongly changes the properties of the spread layers. In this case, the BAM images, unlike Figure 6A–C, show a transition from a heterogeneous spread layer (Figure 6E) to a homogeneous liquid surface. The surface morphology changes in the latter case after a small mechanical disturbance by a thin needle (Figure 6G). This peculiarity indicates, similar to BLG microgel spread layers, a relatively homogeneous particle distribution along the surface just after spreading. The two-fold surface compression decreases the sensitivity of the surface morphology to slight mechanical disturbances, as is observed for spread LYS microgel layers without ethanol (Figure 6F).

The dependences of the dynamic surface elasticity on surface pressure and compression isotherms for LYS microgel layers, which were spread with ethanol (Figure 6F and Figure 8a–c), are similar to those for spread BLG microgel layers. The dynamic surface elasticity increases up to 130 mN/m (Appendix A) and the compression isotherms resemble that of BLG microgel layers (Appendix A). At the same time, the collapse mechanism of LYS microgel particles still differs from that of BLG particles, since the compression isotherms in the former case remain smooth, even at high compressions. Therefore, it is possible to assume that LYS particles are softer than BLG particles, the layer of LYS microgel is less rigid, and its collapse mechanism is similar to that for a layer of poly(N-isopropylacrylamide) (PNIPAM) microgel [44]. While strong compression of a layer of BLG microgel results in the formation of three-dimensional clusters of protein particles and concomitant random fluctuations of the surface pressure [30], the LYS microgel layer contains these three-dimensional clusters even at zero surface pressures, and the layer compression leads only to an increase in its thickness without significant changes to its morphology, and to a smooth compression isotherm.

It is important to note that there is no visible alteration of the aggregate size and shape of the LYS aggregates (Figure 6F), as observed by SEM (Figure 8). At zero surface pressure, there are separate clusters of the aggregates at the surface, but they are mainly some patches not of a multilayer, but of a monolayer. They are coalesced at the beginning of surface compression, and then form a continuous monolayer (Figure 8b,c). The further compression results in the formation of thin wrinkles in the layer (Figure 8a). It is possible to assume that although the aggregate morphology does not change, the ethanol influences the surface of the aggregates and leads to the formation of a relatively soft corona of partly unfolded LYS molecules. In this case, the formation of an aggregate monolayer can be thermodynamically more favorable than a multilayer formation, because of the modification of the corona at the water–air interface and stretching of some unfolded protein chains in the corona along the interface. A similar behavior is described for PNIPAM microgel layers at the same interface [44,45]. The SEM microphotographs show the formation of some bonds between neighboring spherulites in the surface layer (Figure 8b). This behavior differs from that of LYS microgel layers, which were spread without ethanol (Figure 8d,e), when one can observe a strong tendency to aggregation while no monolayer is formed.

## 4. Conclusions

The properties of spread layers of LYS microgel particles on aqueous subphases differ significantly from the properties of the BLG aggregate layer. In the latter case, the spread protein layer is mainly a monolayer and its collapse starts only at relatively high surface pressures, leading to fluctuations in the surface pressure. On the contrary, the formation of tough, three-dimensional clusters of LYS aggregates occurs immediately during the process of spreading, and the protein spherulites do not spread further along the liquid surface. As a result, the liquid surface contains large almost empty regions and some patches of high local concentration of the microgel. The intensity of the amide bands of IRRAS spectra fluctuate strongly in the course of scanning along the liquid surface and, in some regions, the intensity of amide bands exceeds by a few times the values for a native LYS monolayer. The surface pressure and the dynamic surface elasticity are close to zero in this case; however, AFM and BAM show the formation of large surface aggregates of protein nanoparticles, and thereby confirm the strong heterogeneity of the surface layer. The surface pressure and surface elasticity start to increase with the surface compression only when the surface area decreases strongly and the clusters of protein aggregates start to interact. The strong cohesion between the microgel particles leading to the formation of compact three-dimensional clusters at the interface is characteristic only for the LYS microgel but not for the BLG spherulites, indicating that the primary and tertiary structures of this protein influence the organization of microgel layers at the surface of an aqueous subphases. Another factor influencing the structure of spread layers of protein aggregates is the properties of the spreading solvent. The addition of only 20 wt.% of ethanol to the protein aggregate dispersion results in the formation of significantly more homogeneous layers, due to the formation of a soft corona around the microgel particles and the increase in lateral interactions between them. The obtained results indicate that the stabilization mechanism of foam and emulsion films can be different with the application of BLG and LYS nanoparticles. In the former case, the film acquires resistance to mechanical perturbations due to the high surface elasticity at a high surface concentration of nanoparticles. In the latter case, the large clusters of nanoparticles at phase boundaries can hinder a close approach of two approaching interfaces.

## Figures and Tables

**Figure 1 polymers-14-03979-f001:**
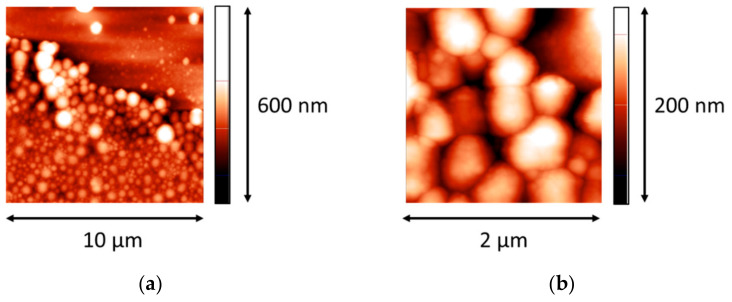
AFM images of a LYS microgel dispersion without purification. (**a**,**b**) images correspond to different magnifications.

**Figure 2 polymers-14-03979-f002:**
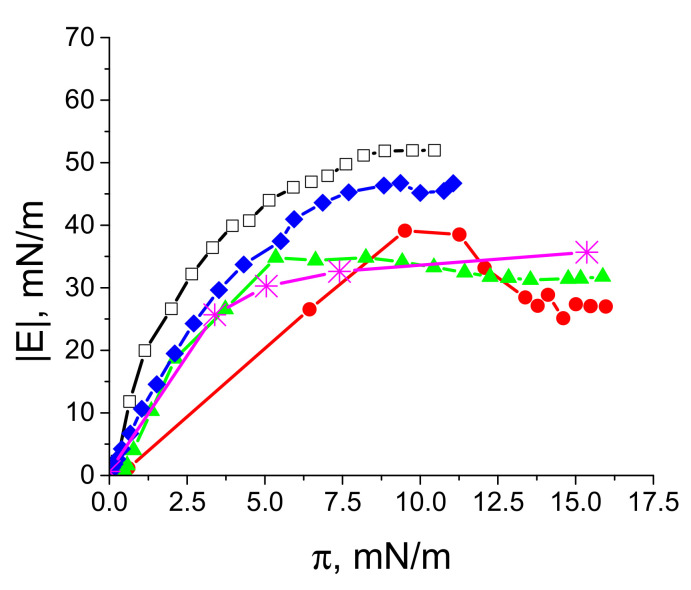
The dependences of the dynamic surface elasticity modulus on surface pressure of spread LYS layers on 0.1 M NaCl solution, pH 10 (open black squares), and for spread layers of a LYS microgel with different numbers of purification cycles: 0 (red circles), 1 (green triangles), 2 (blue diamonds), 3 (magenta snowflakes). The last curve was obtained by surface layer compression after spreading.

**Figure 3 polymers-14-03979-f003:**
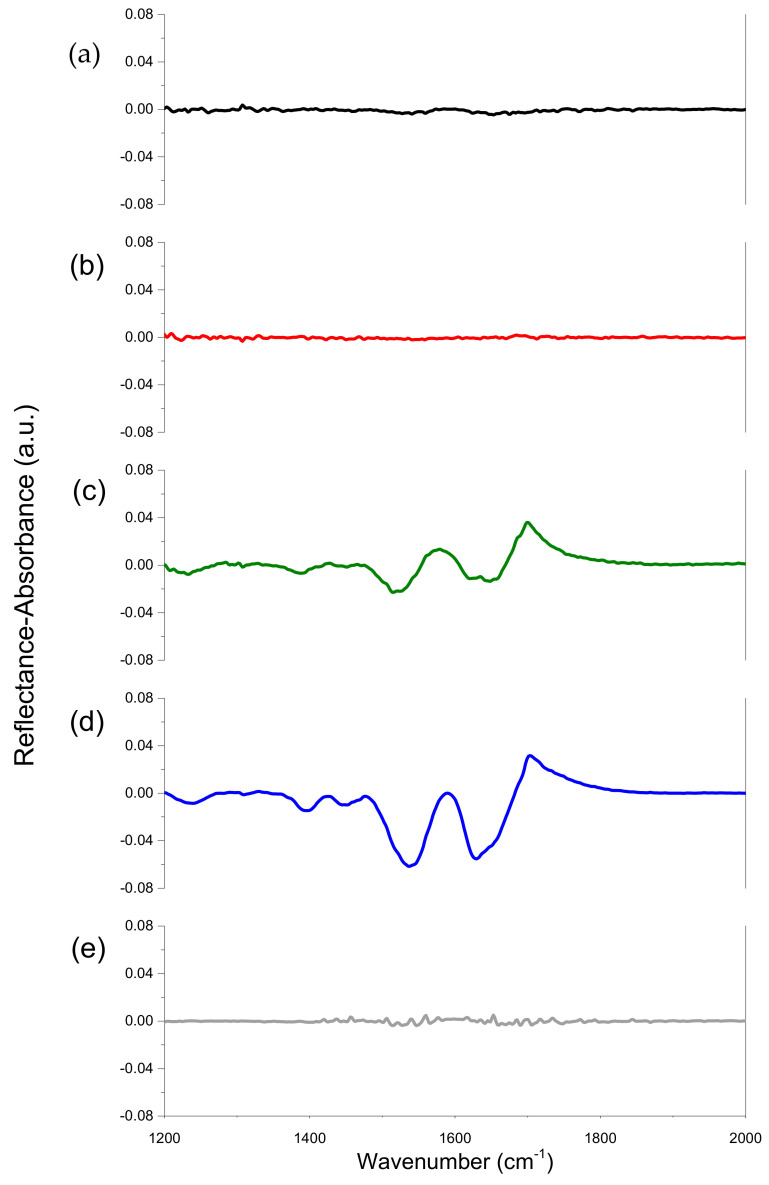
IRRAS spectra of spread layers of native LYS (**a**), LYS microgel (**b**,**c**), BLG microgel (**d**), native BLG (**e**). LYS (native and spherulites) was spread on 0.1 M NaCl solution at pH 10. The (**b**) corresponds to the beam reflection from the surface region far from that where the dispersion is spread and the green line corresponds to the beam reflection from the surface region where the spreading occurs. BLG (native and spherulites) was spread onto 0.1 M NaCl solution at pH 5.5.

**Figure 4 polymers-14-03979-f004:**
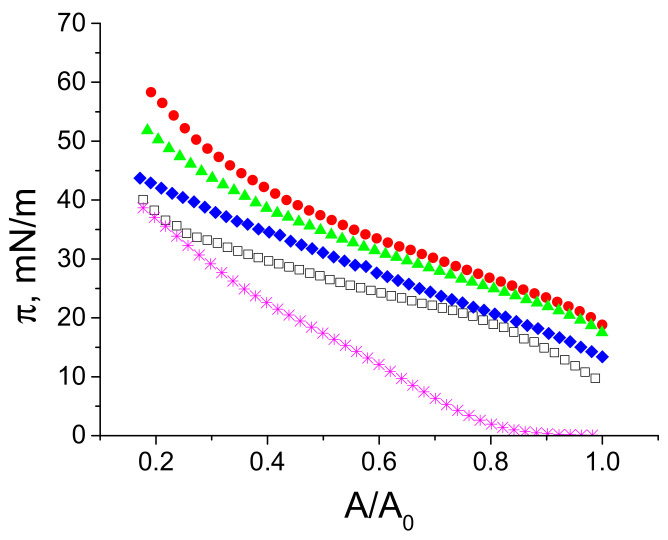
Compression isotherms of spread LYS layers on 0.1 M NaCl solution at pH 10 (open black squares) and the isotherms for spread layers of LYS microgel with different number of purification cycles: 0 (red circles), 1 (green triangles), 2 (blue diamonds), 3 (magenta snowflakes).

**Figure 5 polymers-14-03979-f005:**
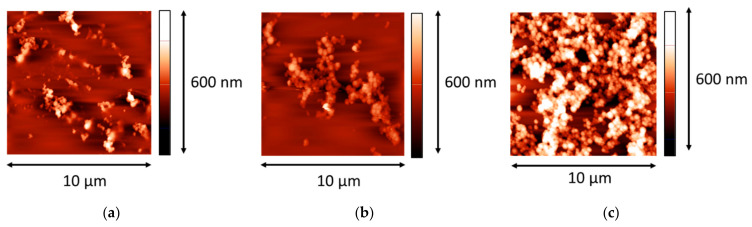
AFM images of spread LYS microgel layers at the surface of 0.1 M NaCl at pH 10. (**a**) corresponds to a dispersion without purification, (**b**,**c**) correspond to dispersions after two and three cycles of purification, respectively. The microgel layers were transferred onto a mica surface after ten-fold compression.

**Figure 6 polymers-14-03979-f006:**
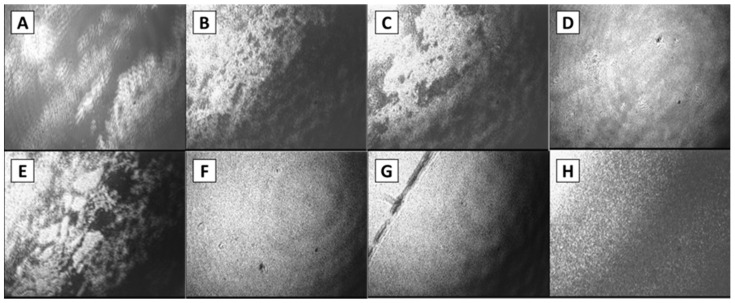
BAM images of spread LYS microgel layers on phosphate buffer. The microgel was purified by a three-cycle procedure of centrifugation and supernatant exchange. The images (**A**–**D**) correspond to layers of LYS microgel without EtOH. The images (**E**–**H**) correspond to layers of LYS microgel with the addition of EtOH to improve spreading. The images (**A**,**B**) and (**E**,**F**) correspond to the increase in surface concentrations by successive additions of the microgel dispersion. The images (**C**,**G**) correspond to layers after mechanical destruction of the layer by a thin needle. The images (**D**,**H**) correspond to two-fold compressed layers after mechanical destruction by a thin needle. The image size is 1300 × 1000 microns.

**Figure 7 polymers-14-03979-f007:**
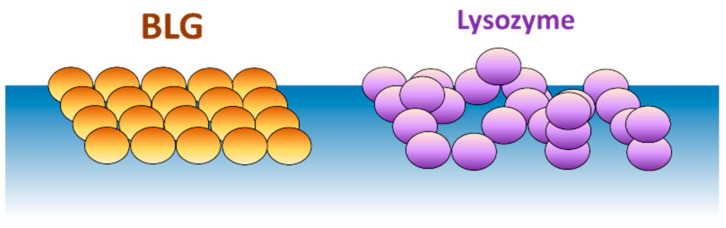
A schematic picture of protein microgel layers.

**Figure 8 polymers-14-03979-f008:**
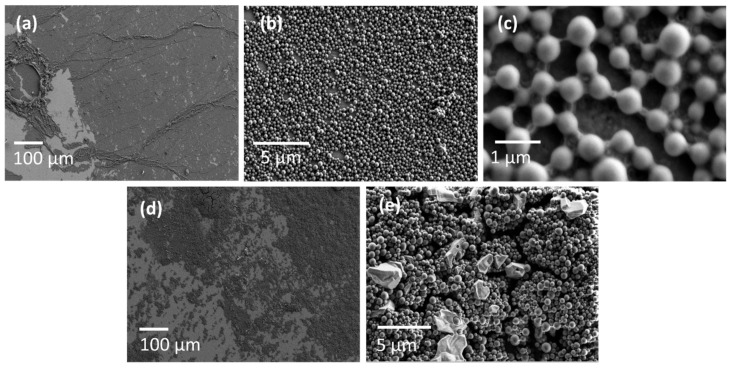
SEM images at different magnifications, (**a**,**d**) ×300, (**b**,**e**) ×10,000, (**c**) ×80,000, of LYS microgel layers, which were spread with the addition of ethanol, at the surface of a phosphate buffer at pH 7 (**a**–**c**) and without the addition of ethanol at the surface of 0.1 M NaCl solution at pH 10 (**d**,**e**).

## Data Availability

Not applicable.

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
