# Peer review of "Spread Layers of Lysozyme Microgel at Liquid Surface"

_polymers, 2022, doi:10.3390/polym14193979_

Round 1
Reviewer 1 Report
This manuscript describes the assembly behavior of lysozyme at air–liquid interfaces, which was characterized by surface dilational rheology, infrared reflection-absorption spectra, Brewster angle microscopy, atomic force microscopy, and scanning electron microscopy. The results were compared with those of ß-lactoglobulin that were previously reported. The findings in this study will contribute to developing protein aggregates as emulsifying and foaming agents in various fields. I basically recommend this manuscript for publication in Polymers. Nevertheless, descriptions on technical terms and experimental methods are insufficient in the current manuscript. Comments are listed below.
1) What is the difference between "aggregates," "microgel", and "spherulites" ? Their detailed descriptions need to be included in their first mention.
2) The preparation methods of lysozyme solutions and aggregates (microgel) need to be described in the experimental section as they are necessary to reproduce this study.
3) Only a mere summary of the results is provided in the conclusion section. The authors should state why the findings in this study are important for developing protein aggregates for interfacial applications.
Author Response
First of all we would like to thank the reviewer for the positive estimation of our manuscript and valuable remarks.
1) What is the difference between "aggregates," "microgel", and "spherulites" ? Their detailed descriptions need to be included in their first mention.
Author’s reply:
The detailed description of the terms "aggregates," "microgel", and "spherulites" is included in the Introduction of the new version of the manuscript:
“Among various protein aggregates, fibrils and dense compact protein nanoparticles have been studied to the most extent [1-6, 9, 10, 12, 17, 19, 23, 24]. The latter ones usually consist of interconnected and partially unfolded protein chains and are known as protein microgel [12, 17, 33, 34]. The LYS and BLG microgels also have an almost spherical shape and therefore protein microgel and protein spherulite are used as synonyms below.”
2) The preparation methods of lysozyme solutions and aggregates (microgel) need to be described in the experimental section as they are necessary to reproduce this study.
Author’s reply:
The preparation methods of lysozyme solutions and aggregates are described now in the new version of the manuscript (first paragraphs of the Materials and Methods section):
“LYS (Sigma-Aldrich, USA, Mw ≈ 14300 Da) and BLG (Sigma-Aldrich, USA, Mw ≈ 18300 Da) were carefully purified before the microgel preparation. Triply distilled water was used to prepare 10 wt % protein solutions. The solution pH was adjusted to 10 by the addition of small amounts of 0.1 M NaOH. After that the solution was dialyzed against water at pH 10 for three days using a cellulose membrane (Sigma-Aldrich, Germany). The protein concentration after dialysis was 2 wt %. To remove undissolved substances the solution was centrifugated (10000 g, 20 min) and filtered through a membrane with a pore size of 200 nm (Vladipore membranes, Russia).
The flask with the purified LYS solution was placed into a hot-water bath at 90°C and heated under rotation for 20-300 min until it became muddy. After that the flask was immediately cooled in an ice-water bath and the obtained microgel dispersion stored in a refrigerator at 4oC.
In order to remove nonreacted protein and peptides the dispersion was centrifuged (at 5000 g for 15 min.) and the supernatant was replaced by water. Shaking a test tube allowed resuspension of microgel particles in water. The whole purification procedure required up to 3 exchange steps.
To spread LYS microgel layers the dispersion was added drop by drop to a liquid subphase using an inclined glass plate, which is partially immersed into the liquid. The subphase was 0.1 M NaCl solution at pH 10, if not otherwise noted. In this case, the high ionic strength and the proximity to the LYS isoelectric point decreased the influence of electrostatic effects. Similar conditions allowed previously to obtain the high elasticity of BLG spread layers [36]. In some cases, as described below, the addition of up to 20 % of ethanol to the spreading dispersion improved the spreading process and allowed obtaining highly homogeneous layers of LYS particles.
Similar procedures, as described in detail previously, were used to obtain BLG microgel and its layers on aqueous subphases [36].”
3) Only a mere summary of the results is provided in the conclusion section. The authors should state why the findings in this study are important for developing protein aggregates for interfacial applications.
Author’s reply:
The importance of the application of protein aggregates is described now in Conclusions (last sentences) of the new version of the manuscript:
“The obtained results indicate that the stabilization mechanism of foam and emulsion films can be different at the application of BLG and LYS nanoparticles. In the former case the film acquires resistance to mechanical perturbations due to the high surface elasticity at high surface concentration of nanoparticles. In the latter case the large clusters of nanoparticles at phase boundaries can hinder a close approach of two approaching interfaces.”
Reviewer 2 Report
In this manuscript, the authors The authors studied the spread layers of lysozyme (LYS) microgel particles at liquid surface. They found that the properties of LYS microgel layers differ significantly from those of ß-lactoglobulin (BLG) microgel layers. The dynamic elasticity of LYS microgel layer does not exceed the values for pure protein layers. There was not specific feature of the layer collapse. The liquid surface contains large empty regions and patches of high local concentration of the microgel particles. The addition of only 20 wt. % of ethanol to the protein aggregate dispersion results in the formation of significantly more homogeneous layers. These results could be published in Polymers after major revision noted below.
1. The experimental detail of preparation of LYS microgels and the spreading microgel suspension was missing.
2. The sources of materials, like LYS and BLG proteins, solvents, were missing.
3. What are the differences between LYS protein and LYS microgels?
4. How to spread the LYS or BLG microgels at the liquid surface? The experimental detail was missing.
5. In the caption of Figure 6, the authors wrote “The microgel was purified by a three-cycles procedure of centrifugation and supernatant exchange.” Why should the microgels be purified? What is the reason for addition of EtOH?
6. What is the difference between LYS spherulites and LYS aggregates?
7. Line 287, the authors wrote “the layer of LYS microgel is less rigid and its collapse mechanism is similar to that for a layer of poly(N-isopropylacrylamide) (PNIPAM) microgel”. What is the collapse mechanism exactly?
Author Response
We would like to thank the reviewer for the positive estimation of our manuscript and valuable remarks.
- The experimental detail of preparation of LYS microgels and the spreading microgel suspension was missing.
Author’s reply:
The experimental detail of preparation of LYS microgels and the spreading microgel suspension are described now in the new version of the manuscript (first paragraphs of the Materials and Methods section):
“LYS (Sigma-Aldrich, USA, Mw ≈ 14300 Da) and BLG (Sigma-Aldrich, USA, Mw ≈ 18300 Da) were carefully purified before the microgel preparation. Triply distilled water was used to prepare 10 wt % protein solutions. The solution pH was adjusted to 10 by the addition of small amounts of 0.1 M NaOH. After that the solution was dialyzed against water at pH 10 for three days using a cellulose membrane (Sigma-Aldrich, Germany). The protein concentration after dialysis was 2 wt %. To remove undissolved substances the solution was centrifugated (10000 g, 20 min) and filtered through a membrane with a pore size of 200 nm (Vladipore membranes, Russia).
The flask with the purified LYS solution was placed into a hot-water bath at 90°C and heated under rotation for 20-300 min until it became muddy. After that the flask was immediately cooled in an ice-water bath and the obtained microgel dispersion stored in a refrigerator at 4oC.
In order to remove nonreacted protein and peptides the dispersion was centrifuged (at 5000 g for 15 min.) and the supernatant was replaced by water. Shaking a test tube allowed resuspension of microgel particles in water. The whole purification procedure required up to 3 exchange steps.
To spread LYS microgel layers the dispersion was added drop by drop to a liquid subphase using an inclined glass plate, which is partially immersed into the liquid. The subphase was 0.1 M NaCl solution at pH 10, if not otherwise noted. In this case, the high ionic strength and the proximity to the LYS isoelectric point decreased the influence of electrostatic effects. Similar conditions allowed previously to obtain the high elasticity of BLG spread layers [36]. In some cases, as described below, the addition of up to 20 % of ethanol to the spreading dispersion improved the spreading process and allowed obtaining highly homogeneous layers of LYS particles.
Similar procedures, as described in detail previously, were used to obtain BLG microgel and its layers on aqueous subphases [36].”
- The sources of materials, like LYS and BLG proteins, solvents, were missing.
Author’s reply:
The sources of materials are described now in the new version of the manuscript (the first paragraph of the Materials and Methods section):
“LYS (Sigma-Aldrich, USA, Mw ≈ 14300 Da) and BLG (Sigma-Aldrich, USA, Mw ≈ 18300 Da) were carefully purified before the microgel preparation. Triply distilled water was used to prepare 10 wt % protein solutions. The solution pH was adjusted to 10 by the addition of small amounts of 0.1 M NaOH. After that the solution was dialyzed against water at pH 10 for three days using a cellulose membrane (Sigma-Aldrich, Germany). The protein concentration after dialysis was 2 wt %. To remove undissolved substances the solution was centrifugated (10000 g, 20 min) and filtered through a membrane with a pore size of 200 nm (Vladipore membranes, Russia).”
- What are the differences between LYS protein and LYS microgels?
Author’s reply:
In our study we used both solutions of protein molecules and dispersions of specially prepared protein microgel. The preparation of LYS microgel is described now in details in the Materials and Methods section of the new version of the manuscript (please cf. the reply to question 1). The preparation of the LYS protein solutions is also described now in this section.
- How to spread the LYS or BLG microgels at the liquid surface? The experimental detail was missing.
Author’s reply:
To spread LYS microgel layers the dispersion was added drop by drop to a liquid subphase using an inclined glass plate, which is partially immersed into the liquid. This sentence was added to the Materials and Methods section. The same method was applied to spread BLG microgel.
- In the caption of Figure 6, the authors wrote “The microgel was purified by a three-cycles procedure of centrifugation and supernatant exchange.” Why should the microgels be purified? What is the reason for addition of EtOH?
Author’s reply:
The purification of the microgel is necessary to remove nonreacted protein and peptides of low molecular weight. This is described now in the new version of the manuscript (Materials and Methods section):
“In order to remove nonreacted protein and peptides the dispersion was centrifuged (at 5000 g for 15 min.) and the supernatant was replaced by water. Shaking a test tube allowed resuspension of microgel particles in water. The whole purification procedure required up to 3 exchange steps.”
The addition of ETOH is necessary to improve the spreading process. The corresponding explanation is added to the caption of Figure 6:
“The images (A-D) correspond to layers of LYS microgel without EtOH. The images (E-H) correspond to layers of LYS microgel with the addition of EtOH to improve spreading (cf. discussion of Fig. 8).”
- What is the difference between LYS spherulites and LYS aggregates?
Author’s reply:
The difference between LYS spherulites and LYS aggregates is now described in the Introduction of the new version of the manuscript:
“Among various protein aggregates, fibrils and dense compact protein nanoparticles have been studied to the most extent [1-6, 9, 10, 12, 17, 19, 23, 24]. The latter ones usually consist of interconnected and partially unfolded protein chains and are known as protein microgel [12, 17, 33, 34]. The LYS and BLG microgels also have an almost spherical shape and therefore protein microgel and protein spherulite are used as synonyms below.”
- Line 287, the authors wrote “the layer of LYS microgel is less rigid and its collapse mechanism is similar to that for a layer of poly(N-isopropylacrylamide) (PNIPAM) microgel”. What is the collapse mechanism exactly?
Author’s reply:
The collapse mechanism of LYS microgel and its difference from the mechanism of BLG microgel is now described in the new version of the manuscript:
“While strong compression of a layer of BLG microgel results in the formation of three-dimensional clusters of protein particles and concomitant random fluctuations of the surface pressure [36], the LYS microgel layer contains these three-dimensional clusters even at zero surface pressures and the layer compression leads only to an increase of its thickness without significant changes of its morphology, and to a smooth compression isotherm.”
Round 2
Reviewer 2 Report
The authors have effectively addressed my comments. I would like to recommend the publication of the revised manuscript in Polymers as is.